# PepRePs: Peptide-Retargeted Phosphatases via Generative Language Models

## Abstract

Phosphorylation encodes a fast, reversible layer of regulation that directly governs protein activity, signaling flow, and cell state transitions. While kinases and phosphatases collectively shape these landscapes, existing tools lack the ability to selectively rewrite phosphorylation states of individual, often disordered, proteins inside cells. Here, we introduce **Pep**tide-**Re**targeted **P**hosphatases (**PepRePs**), a genetically encodable proteome-editing platform that uses protein language model-derived peptide binders to localize phosphatase activity to user-specified targets. PepRePs enable targeted dephosphorylation of endogenous and exogenous proteins, including Rab8a and tau, reducing site-specific phosphorylation and producing downstream functional effects linked to altered cellular behavior. By coupling programmable peptide binders with modular phosphatase domains, PepRePs provide a mechanism-aware way to perturb signaling networks at the level of post-translational control rather than gene expression. This work positions targeted dephosphorylation as a scalable strategy for probing and reprogramming phosphorylation-driven cell states, with implications for both systems biology and therapeutic intervention in diseases dominated by signaling dysregulation.

## 1 Introduction

Cell state is governed not only by gene expression but by rapid, reversible post-translational regulation of protein activity, localization, and interaction networks. Among post-translational modifications (PTMs), phosphorylation plays a central role, acting as a dynamic regulatory layer that directly modulates signaling flux, transcriptional programs, and phenotypic transitions (Khoury et al., 2011; Wu et al., 2021; Singh et al., 2017; Fischer, 2013). Addition or removal of phosphate groups on serine, threonine, and tyrosine residues induces conformational and electrostatic changes that alter protein function, assembly, and stability, thereby reshaping downstream cellular behavior (Fischer, 2013; Singh et al., 2017). Dysregulated phosphorylation is causally linked to cancer, neurodegenerative disorders such as Alzheimer's and Huntington's disease, and developmental pathologies (Singh et al., 2017; Khoury et al., 2011). Despite its centrality to cell state control, phosphorylation remains difficult to manipulate with precision at the level of individual proteins inside cells.

Most existing approaches perturb phosphorylation indirectly by inhibiting or activating kinases and phosphatases, broadly reshaping signaling networks (Ferguson & Gray, 2018; Vintonyak et al., 2009). More targeted strategies have recently emerged. Phosphorylation-targeting chimeras (PhosTACs) use bifunctional small molecules to recruit endogenous phosphatases to specific substrates (Chen et al., 2021), while peptide-based dephosphorylation targeting chimeras (DEPTACs) leverage short binding motifs to recruit phosphatase activity to disease-relevant proteins such as tau (Xiao et al., 2024). In parallel, genetically encodable proteome-editing systems have demonstrated that short peptide binders can act as modular intracellular targeting elements. Peptide-guided ubiquitination systems (ubiquibodies, uAbs) recruit E3 ligases to induce selective degradation (Portnoff et al., 2014; Bhat et al., 2025; Brixi et al., 2023; Chen et al., 2025a), while peptide-guided deubiquitinases (deubiquibodies, duAbs) enable targeted protein stabilization (Hong et al., 2025). Together, these approaches establish a general design principle: compact peptide binders can redirect enzymatic activity toward specific substrates without reliance on small-molecule binding pockets or stable tertiary structure.

Scaling such proteome-editing strategies requires programmable peptide binders to diverse, often disordered targets that are inaccessible to structure-based methods such as RFdiffusion, BindCraft, or BoltzGen (Watson et al., 2023; Pacesa et al., 2025; Stark et al., 2025). Protein language models (pLMs) address this bottleneck by learning sequence-level representations without explicit structural inputs (Lin et al., 2023; Elnaggar et al., 2021). Recent approaches such as SaLT&PepPr, PepPrCLIP, and PepMLM established that short peptide binders can be generated and prioritized from sequence alone (Brixi et al., 2023; Bhat et al., 2025; Chen et al., 2025a), and can be used for the design of potent uAbs and duAbs (Hong et al., 2025). Even newer methods, including moPPIt and SOAPIA, extend this capability by

introducing motif-level control and isoform-aware selectivity, enabling increasingly precise targeting within complex proteomes (Chen et al., 2025b; Vincoff et al., 2025a).

While peptide-guided ubiquitination and deubiquitination systems (uAbs and duAbs) regulate protein abundance, phosphorylation governs protein function on faster timescales and often dictates signaling state without altering expression levels (Khoury et al., 2011; Srividhya et al., 2007). Here, we introduce **Pep**tide-**Re**targeted **P**hosphatases (**PepRePs**), a genetically encodable platform that fuses pLM-designed peptide binders to serine/threonine phosphatase subunits, enabling selective dephosphorylation of endogenous proteins. We show that PepRePs can be programmed to dephosphorylate disease-relevant targets such as Rab8a and tau (Shi et al., 2017; Zhou et al., 2008), producing downstream functional effects and linking precise manipulation of phosphorylation state to changes in cellular behavior.

## 2 METHODS

**Binder Design** The TB1, TB2, and TB3 peptides were derived experimentally as described in literature (Xiao et al., 2024). Novel binding peptides designed in this study - including Rab8a-SnP-1 and Rab8a-SnP-2 - were generated by the previously described SaLT&PepPr (Brixi et al., 2023) (`https://huggingface.co/ubiquitx/saltnpeppr`), PepMLM (Chen et al., 2025a) (`https://huggingface.co/ChatterjeeLab/PepMLM-650M`), and moPPIt algorithms (Chen et al., 2025b) (`https://huggingface.co/ChatterjeeLab/moPPIt`). All binder sequences can be found in Supplementary Table A2.

**Generation of Plasmids** All PepRePs plasmids were generated from the standard pcDNA3 vector, harboring a cytomegalovirus (CMV) promoter. An Esp3I restriction site was introduced immediately upstream of the phosphatase subunits (PP1c and PP2Aa) and flexible GSGSG linker via Gibson Assembly using synthesized gene fragments (Azenta). For target-specific PepRePs assembly, oligos for candidate peptides were annealed and ligated via T4 DNA Ligase into the Esp3I-digested duAb backbone. Assembled constructs were transformed into 50 $\mu$L NEB Turbo Competent *Escherichia coli* cells, and plated onto LB agar supplemented with the appropriate antibiotic for subsequent sequence verification of colonies and plasmid purification (Azenta).

**Cell Culture** The HEK293T and HEK293 Flp-In cells were maintained in Dulbecco's Modified Eagle's Medium (DMEM) supplemented with 100 units/mL penicillin, 100 mg/mL streptomycin, and 10% fetal bovine serum (FBS). For endogenous target screening in cell lines, pcDNA-PepRePs (750 ng) plasmids were transfected into cells as triplicates ($3 \cdot 10^4$/well in an 8-well chamber) with Lipofectamine 2000 in Opti-MEM. Cells were harvested after 48 hours for subsequent cell fractionation and immunoblotting.

**Tau seeding assay** Tau-aggregated seeds were prepared by incubating tau 2N4R p301S monomers in 20 $\mu$M heparin, 2 mM DTT and 1X protease inhibitors in PBS for 72 hours at 37C while shaking at 220 rpm. Prior to seeding, HEK 293 Flp-In T-Rex cells expressing tau variants were plated for 24 hours prior to tetracycline (2 $\mu$g/mL) treatment to induce Halo-tau variants expression. Aggregated seeds were mixed with tau-seeding reagent composed of Lipofectamine 2000, Opti-MEM, and tetracycline prior to cell treatment for 48 hours. HEK 293 Flip-IN cells were transfected with PepRePs with Lipofectamine 2000 in Opti-MEM for 72 hours. Inhibitor-specfic treatment cells were treated with 10 nM okadaic acid (OA) 16 hours prior to collection. The medium was removed 72 hours post-transfection followed by cell fixation using 4% paraformaldehyde solution (BD Biosciences) at 4C for 20 minutes. After fixation, the samples were immersed in PBS and stored at 4C until confocal microscopy analysis.

**Halotrap pulldown** HEK293T cells were co-transfected with constructs HA-Rab8a, Myc-LKRR2 and PepREPs with Opti-MEM and Lipofectamine 2000 for 48 hours prior to harvest. Cells were collected and lysed in lysis buffer, which contains 150 mM NaCl, 1 mM EDTA, 25 mM Tris-HcL, 1% (v/v) NP40, phosphatase inhibitor (PhosStop, Roche), and protease inhibitor (cOmplete, Roche). The supernatants were collected after spinning the samples at 15000 rpm for 20 minutes at 4C. Following a 2 hour incubation period with HaloLink beads, the samples were eluted with 100 mM citric acid (pH 3.0) and pipetted up and down for 1 minute at room temperature. Samples were further spun down at 2500 g for 5 minutes at 4C then neutralized with 1 M Tris (pH 10.4), and further rounds of elution were completed to enrich all potential interacting proteins. Final samples were used for immunoblotting, and will further be submitted for mass spectrometry analysis in future work.

**Immunoblotting** On the day of harvest, cells were detached were washed twice with ice-cold 1X PBS. Cell lysis and immunoblotting were performed according to standard protocols. Proteins were probed with mouse anti-tau antibody (Santa Cruz, sc-21796), mouse anti-pT181 tau antibody (ThermoFisher, MN1050), rabbit anti-Rab8a antibody

(Cell Signaling Technology, 6975), rabbit anti-pT72 Rab8a (Abcam, MJF-R20), or mouse anti-GAPDH (Millipore, MAB374) overnight at 4C. The blots were washed and probed with secondary antibodies for 1-2 hours at room temperature before imaging. Densitometry analysis of protein bands in immunoblots was performed using FIJI software as described here: (https://imagej.nih.gov/ij/docs/examples/dot-blot/).

**Statistical Analysis and Reproducibility**  Sample sizes were not predetermined based on statistical methods but were chosen according to the standards of the field (three independent biological replicates for each condition), which gave sufficient statistics for the effect sizes of interest. All data were reported as average values with error bars representing standard deviation (SD). For individual samples, statistical analysis was performed using the two-tailed Student's t test using GraphPad Prism 10 software, with calculated p values are represented as follows: *, $p < 0.05$, **, $p < 0.01$. No data were excluded from the analyses. The experiments were not randomized. The investigators were not blinded to allocation during experiments and outcome assessment.

## 3 RESULTS

### 3.1 ENGINEERING PEPTIDE-RETARGETED PHOSPHATASES (PEPREPS)

Previous work has established that short peptide binders can be fused to enzymatic domains to enable targeted protein degradation (Portnoff et al., 2014) or stabilization (Hong et al., 2025). Building on this paradigm, we designed a programmable, peptide-guided dephosphorylation system (PepRePs) by fusing target-binding peptides to serine/threonine phosphatase catalytic subunits (Figure 1A). As an initial test case, we focused on Rab8a, a membrane-trafficking GT-Pase whose phosphorylation state is implicated in neurodegenerative disorders including Parkinson's and Alzheimer's disease (Gitler et al., 2008; Malik et al., 2021; Shi et al., 2017; Wang et al., 2013). Candidate peptide binders to Rab8a were selected using SaLT&PepPr based on known interaction partners (Supplementary Table A2) and fused to multiple phosphatase catalytic subunits (Supplementary Table A1).

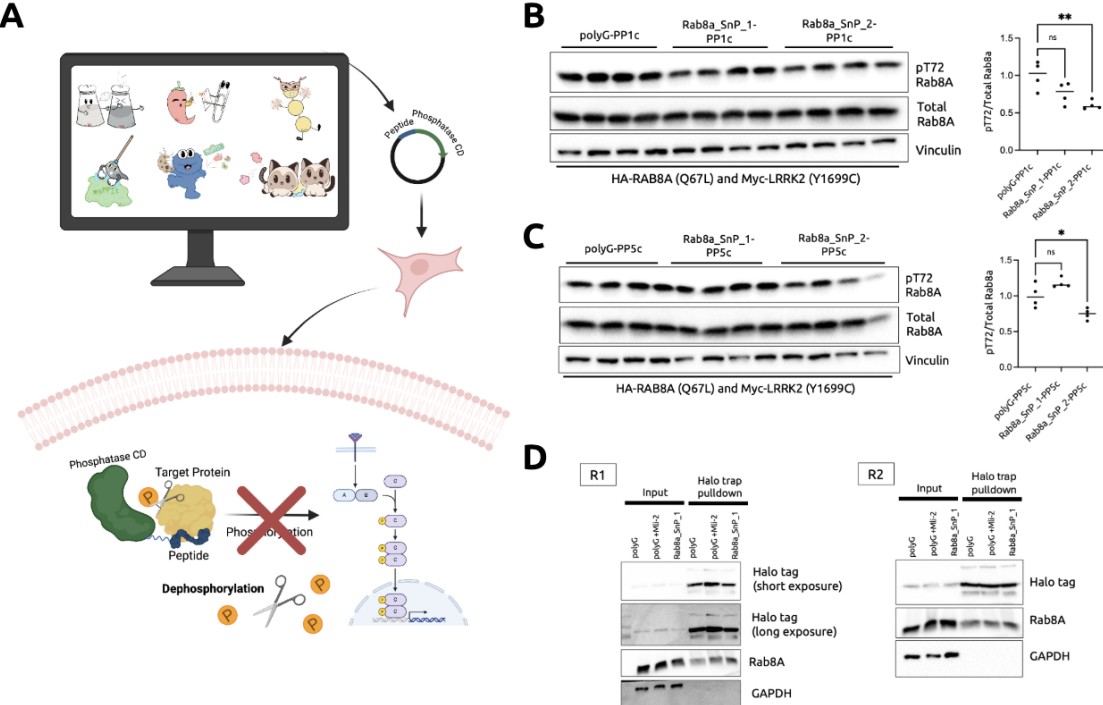

Figure 1: **PepRePs architecture engineering.** A) Genetically-encodable architecture for peptide-recruiting phosphatases. B) Rab8a-targeting PepRePs employing the PP1 catalytic subunit (PP1c). C) Rab8a-targeting PepRePs employing the PP5 catalytic subunit (PP5c). D) Halotrap pulldown of RAb8a (replicates 1 and 2, denoted as R1 and R2, respectively) showing differential expression of Rab8a Thr72 phosphorylation due to PepRePs treatment.

Among the phosphatase variants tested, PP1c-based PepRePs produced the most robust reduction in Rab8a Thr72 phosphorylation, as measured by phospho-specific immunoblotting, compared to other catalytic subunit candidates (Figure 1B,C; Supplementary Table A1) (Malik et al., 2021). To independently validate this effect, we performed Halo-trap pulldown assays (Hu et al., 2023) using Halo–$\alpha$-synuclein to enrich Rab8a complexes (Yin et al., 2014). Consistent with whole-cell immunoblotting, Halo-trap pulldown followed by phospho-specific detection confirmed reduced Rab8a Thr72 phosphorylation upon expression of the Rab8a-SnP-2–PP1c PepReP construct (Figure 1D).

## 3.2 PROGRAMMABILITY OF PEPREPS ACROSS DISTINCT TARGETS

To assess the programmability of PepRePs beyond Rab8a, we next engineered constructs using experimentally validated tau-targeting peptides. The TB1 peptide (Supplementary Table A2) was previously developed for tau dephosphorylation in Alzheimer's disease mouse models (Zheng et al., 2021), while TB2 and TB3 have been shown to recognize tau fibrils and reduce phosphorylated tau levels in cellular and in vivo contexts (Xiao et al., 2024). We focused on the human 2N4R tau isoform, which exhibits a high propensity for aggregation in the brain (Xiao et al., 2024; Martin et al., 2013) (Figure 2A).

In contrast to the Rab8a case, PepRePs incorporating the PP2A regulatory subunit (PP2Aa) produced the most pronounced reduction in tau pT231 phosphorylation (Capano et al., 2022) (Figure 2B). This observation is consistent with prior studies demonstrating that PP2A is a major tau-directed phosphatase in the brain (Zhou et al., 2008; Martin et al., 2013; Qian et al., 2010), and highlights the ability to pair distinct peptide binders with different phosphatase domains to tune target-specific dephosphorylation. Importantly, PP2Aa-based PepRePs enabled substantial reduction of tau aggregation in a cellular tau seeding assay performed in HEK293 Flp-In cells (Hu et al., 2023) (Figure 2C).

To further confirm that these effects were driven by PepRePs-mediated dephosphorylation, cells were treated with the phosphatase inhibitor okadaic acid (OA), which preserved tau phosphorylation and aggregate formation despite PepRePs expression (Hu et al., 2023) (Figure 2D). Together, these results demonstrate that PepRePs can be readily reprogrammed across distinct targets and phosphatase subunits, and that targeted dephosphorylation translates into measurable downstream cellular phenotypes relevant to disease-associated signaling states.

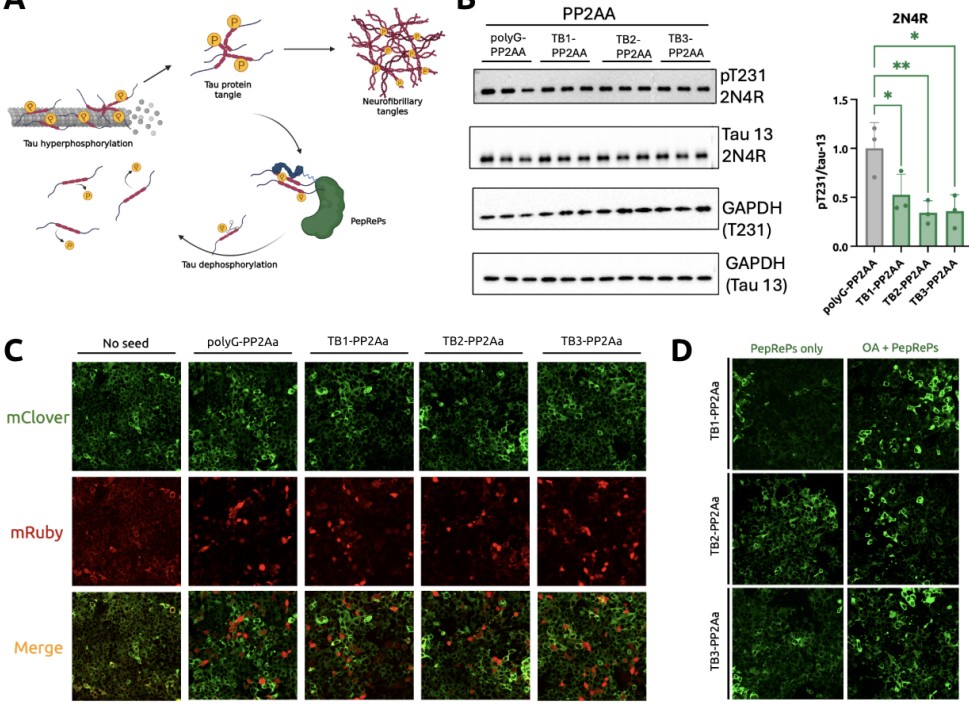

Figure 2: **Extending PepRePs programmability.** A) Tau dephosphorylation and decumulation of neurofibrillary tangles. B) Tau pT231 2N4R dephosphorylation in tau-induced HEK293 Flp-In Trex cells. C) Confocal microscopy imaging showing significant puncta reduction upon transfection of tau-targeting PepRePs. D) Confocal microscopy imaging showing PepRePs-dependent puncta reduction via okadaic acid (OA) treatment.

## 4   CONCLUSION

In this work, we demonstrate that peptide-retargeted phosphatases (PepRePs) provide a modular and genetically encodable strategy for perturbing phosphorylation states of specific intracellular proteins, including targets that are otherwise inaccessible to small molecule-based approaches. By fusing short, programmable peptide binders to phosphatase subunits, PepRePs enable targeted dephosphorylation without globally perturbing kinase or phosphatase activity. Integrated with rapid peptide binder generation pipelines (Brixi et al., 2023; Chen et al., 2025a;b), our results highlight the simplicity and adaptability of this architecture, showing that PepRePs can be readily reprogrammed across distinct targets to induce measurable downstream functional effects.

Beyond individual targets, PepRePs offer a route toward systematic exploration of phosphorylation-driven cell state regulation. Phosphorylation acts as a fast, reversible control layer that governs signaling flow, protein function, and phenotypic transitions (Wu et al., 2021; Khoury et al., 2011). By enabling selective rewriting of phosphorylation states at the level of individual proteins or motifs, PepRePs provide a means to probe how local post-translational perturbations propagate through signaling networks to reshape global cellular behavior. As a genetically encoded modality (Portnoff et al., 2014; Hong et al., 2025), PepRePs can be deployed across diverse cellular contexts, enabling controlled perturbation experiments that map causal relationships between phosphorylation patterns and emergent cell states, rather than relying on correlative readouts alone.

Looking forward, coupling PepRePs with language model–generated peptide binders (Bhat et al., 2025; Brixi et al., 2023; Chen et al., 2025a) enables a scalable framework for interrogating previously unexplored regions of the phosphoproteome. Peptide guides can be further refined to selectively bind post-translationally modified or mutant protein isoforms (Peng et al., 2025; Vincoff et al., 2025b;a; Chen et al., 2025b), allowing increasingly precise control over state-specific signaling nodes. More broadly, the PepRePs architecture is not limited to phosphatases and can be extended to other PTM-modifying enzymes, including acetylases, glycosylases, and kinases, supporting a unified, programmable approach to post-translational regulation. Together, this work positions PepRePs as a general proteome editing platform that bridges generative modeling and experimental perturbation, enabling principled manipulation of post-translational state space with implications for both therapeutic development and foundational studies of cellular decision-making.

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

# A   SUPPLEMENTARY TABLES

Table A1: Phosphatase subunit and linker sequences used in this study.

| Label | Type | Amino Acid Sequence | UniProt ID |
|---|---|---|---|
| L1 | Linker | GSGSG | N/A |
| PP1c | Phosphatase catalytic subunit | MSDSEKLNLDSIIGRLLEVQGSRPGKNVQLTENEIRGLCLKSRE IFLSQPILLELEAPLKICGDIHGQYYDLLRLFEYGGFPPESNYLF LGDYVDRGKQSLETICLLLAYKIKYPENFFLLRGNHECASINRI YGFYDECKRRYNIKLWKTFTDCFNCLPIAAIVDEKIFCCHGGL SPDLQSMEQIRRIMRPTDVPDQGLLCDLLWSDPDKDVQGWGE NDRGVSFTFGAEVVAKFLHKHDLDLICRAHQVVEDGYEFFAK RQLVTLFSAPNYCGEFDNAGAMMSVDETLMCSFQILKPADKN KGKYGQFSGLNPGGRPITPPRNSAKAKK | P62136 |
| PP2Aa | Phosphatase regulatory subunit | MAAADGDDSLYPIAVLIDELRNEDVQLRLNSIKKLSTIALALG VERTRSELLPFLTDTIYDEDEVLLALAEQLGTFTTLVGGPEYV HCLLPPLESLATVEETVVRDKAVESLRAISHEHSPSDLEAHFVP LVKRLAGGDWFTSRTSACGLFSVCYPRVSSAVKAELRQYFRN LCSDDTPMVRRAAASKLGEFAKVLELDNVKSEIIPMFSNLASD EQDSVRLLAVEACVNIAQLLPQEDLEALVMPTLRQAAEDKSW RVRYMVADKFTELQKAVGPEITKTDLVPAFQNLMKDCEAEV RAAASHKVKEFCENLSADCRENVIMSQILPCIKELVSDANQHV KSALASVIMGLSPILGKDNTIEHLLPLFLAQLKDECPEVRLNIIS NLDCVNEVIGIRQLSQSLLPAIVELAEDAKWRVRLAIIEYMPLL AGQLGVEFFDEKLNSLCMAWLVDHVYAIREAATSNLKKLVE KFGKEWAHATIIPKVLAMSGDPNYLHRMTTLFCINVLSEVCG QDITTKHMLPTVLRMAGDPVANVRFNVAKSLQKIGPILDNSTL QSEVKPILEKLTQDQDVDVKYFAQEALTVLSLA | P30153 |
| PP5c | Phosphatase catalytic subunit | GKVTISFMKELMQWYKDQKKLHRKCAYQILVQVKEVLSKLS TLVETTLKETEKITVCGDTHGQFYDLLNIFELNGLPSETNPYIF NGDFVDRGSFSVEVILTLFGFKLLYPDHFHLLRGNHETDNMN QIYGFEGEVKAKYTAQMYELFSEVFEWLPLAQVINGKVLIMH GGLFSEDGVTLDDIRKIERNRQPPDSGPMCDLLWSDPQPQNGR SISKRGVSCQFGPDVTKAFLEENNLDYIIRSHEVKAEGYEVAH GGRCVTVFSAPNYCDQMGNKASYIHLQGSDLRRPQFHQFTAV PHPNVKPMAYANTLLQLGMM | P53041 |

Table A2: Peptide binders used in this study.

| Binder Name | Target UniProt ID | Target Name | Binder Sequence | Derivation Source |
|---|---|---|---|---|
| TB1 | P10636-8 | 2N4R Tau | YQQYQDATADEQG | Zheng et al. |
| TB2 | P10636-8 | 2N4R Tau | TLKIVW | Xiao et al. |
| TB3 | P10636-8 | 2N4R Tau | DVVMINKKRK | Xiao et al. |
| Rab8a_SnP_1 | P61006 | Rab8a | DDEKEQFLYHLLSFNAV | SaLT&PepPr |
| Rab8a_SnP_2 | P61006 | Rab8a | LGQELEELTASLFEEAHKMV | SaLT&PepPr |

