# OpenReview forum: "PepRePs: Peptide-Retargeted Phosphatases via Generative Language Models"
_ICLR.cc/2026/Workshop/FM4Science — ICLR 2026 Workshop FM4Science Poster_

### Official Review · Reviewer_cHxq · 2026-02-15
**This submission introduces Peptide-Retargeted Phosphatases (PepRePs), a genetically encodable and modular platform for the site-specific dephosphorylation of endogenous proteins. The core innovation is the use of protein language models (pLMs) to design compact peptide binders that act as targeting elements, recruiting phosphatase catalytic subunits to user-specified substrates such as Rab8a and tau. Unlike existing approaches like PhosTACs, which rely on bifunctional small molecules, PepRePs are programmable via sequence alone, making them suitable for targeting disordered proteins that lack traditional binding pockets. The work demonstrates successful dephosphorylation and a reduction in tau aggregation in cellular models, positioning PepRePs as a scalable tool for both foundational biology and therapeutic discovery**

**Rating:** 8
**Confidence:** 4

**Review:**

Overall, this work is of high quality. It demonstrates significant potential to probe the causal relationships between specific phosphorylation events and global cellular behavior. By enabling the selective dephosphorylation of motifs on proteins like tau—which is implicated in Alzheimer's and other tauopathies—the authors provide a mechanism for investigating signaling flux without the global disruption typically caused by kinase or phosphatase inhibitors.

The finding that different targets require specific phosphatase subunits (e.g., PP1c for Rab8a vs. PP2Aa for tau) highlights the necessity of the modularity inherent in the PepRePs design.

## Strengths

1. Genetically Encodable Modularity: The system is entirely protein-based, allowing for endogenous expression and eliminating the stoichiometric limitations of occupancy-driven small-molecule inhibitors.
2. Structure-Agnostic Targeting: By leveraging pLMs trained on evolutionary-scale sequence data, the model can design binders for disordered proteins that structural methods like *RFdiffusion* or *BoltzGen* might struggle with.
3. Broad Programmability: The authors successfully validated the system across multiple targets and phosphatase subunits, demonstrating that the architecture can be readily adapted to diverse signaling nodes.
4. Functional Validation: The study goes beyond dephosphorylation to connect findings to a disease-relevant phenotype—the reduction of tau aggregation—using a seeding assay in HEK293 cells.
5. Mechanistic Clarity: The use of okadaic acid as a control confirms that the observed effects are indeed due to the recruited phosphatase activity, providing a robust experimental safeguard.

## Weaknesses

- Subunit Pairing Sensitivity: The requirement for target-specific screening of phosphatase subunits (e.g., the failure of PP1c to dephosphorylate tau as effectively as PP2Aa) suggests that the "rules" of recruitment are not yet fully predictable by the current FM framework.
- In-vivo Delivery Hurdles: As a genetically encoded system, eventual therapeutic application will require efficient gene delivery mechanisms (e.g., AAVs or mRNA-LNPs), which are inherently more complex than small-molecule delivery.
- Potential Off-target Dephosphorylation: While the peptide binders are designed for high affinity, the study lacks a comprehensive mass-spectrometry-based phosphoproteomic analysis to rule out global perturbations.

---

### Official Review · Reviewer_cQo8 · 2026-02-18
**The paper introduces Peptide-Retargeted Phosphatases (PepRePs), a genetically encodable platform for targeted dephosphorylation of proteins using peptide binders derived from protein language models (pLMs) like SaLT&PepPr, PepMLM, and moPPIt. PepRePs fuse these binders to phosphatase subunits (e.g., PP1c, PP2Aa) to localize enzymatic activity to specific targets, such as Rab8a and tau, which are implicated in neurodegenerative diseases. Experiments in HEK293T cells demonstrate reduced site-specific phosphorylation (e.g., Rab8a Thr72, tau pT231) and downstream effects like decreased tau aggregation. The work positions PepRePs as a scalable tool for perturbing signaling networks at the post-translational level, with potential applications in systems biology and therapeutics.**

**Rating:** 5
**Confidence:** 3

**Review:**

1) Quality

Proof-of-concept with construct design and multiple readouts (phospho-immunoblotting, pulldown validation, microscopy phenotype, inhibitor rescue). However, it remains an early demonstration across a small set of targets/sites and largely HEK-derived systems.

2) Clarity

High-level motivation and system concept are clear, and figures support the narrative. Some methodological and interpretation gaps remain, e.g., specificity/off-target dephosphorylation characterization, quantitative binder properties, and clearer rationale for phosphatase subunit choices per target.

3) Originality

The key novelty is a genetically encoded, peptide-guided dephosphorylation strategy that is coupled to pLM-derived peptide binders.  It also extends prior “peptide-guided enzyme retargeting” paradigms (uAbs/duAbs) into phosphorylation control.

4) Significance

Potentially impactful for mechanistic signaling studies and disease contexts where phosphorylation state (vs abundance) is causal. Translational significance is plausible but not yet established due to limited scope and incomplete off-target/safety profiling.

5) Pros

* Modular, genetically encodable architecture for targeted dephosphorylation.
* Demonstrated across two distinct targets (Rab8a, tau) with site-specific phospho readouts.
* Mechanistic support via inhibitor (okadaic acid) reversing phenotype.
* Uses pLM-based peptide generation/selection to scale targeting beyond structured proteins.
* Shows downstream cellular consequences (tau aggregation reduction), not just biochemical changes.
* Clear path to extensibility to other PTM enzymes (conceptually).

6) Cons

* AI/FM aspects not as comprehensive as the other submissions that the reviewer has reviewed so far.
* Limited target/site breadth.
* Generality across the phosphoproteome not yet proven.
* Off-target effects and global phosphoproteome impact are not quantified (no phosphoproteomics).
* Binder properties (affinity/specificity, competition, isoform selectivity) not deeply characterized.
* Mostly overexpression/HEK-based assays; endogenous physiological validation is limited.
* Experimental rigor notes: no randomization/blinding; sample-size justification is field-standard but modest.
* Some “future work” items (e.g., mass spec follow-up) indicate the story is not fully closed.

Question for PC: Do we have an option for Posters please?  Thanks!

---

### Meta-Review · Area_Chair_k4FJ · 2026-02-28

**Recommendation:** Accept (Poster)
**Confidence:** 3

**Metareview:**

This work describes a proof of concept of PepRePs, a genetically encodable proteome-editing platform that uses protein language model-derived peptide binders to localize phosphatase activity to user-specified targets. Reviewers agree that the approach has the potential to be significant, particularly for "prob[ing] the causal relationships between specific phosphorylation events and global cellular behavior". While the foundation models aspects are less comprehensive, protein language models are used effectively to facilitate structure-agnostic targeting and engagement of disordered proteins. While some reviewers note potential methodological gaps that could challenge the generality and downstream application of the platform, they agree that the work in its current form is of interest to the workshop.

---

### Decision · Program_Chairs · 2026-03-03

Accept (Poster)